# Mdm2-Mediated Ubiquitination Plays a Pivotal Role in Differentiating the Endocytic Roles of GRK2 and Arrestin3

**DOI:** 10.3390/ijms26073238

**Published:** 2025-03-31

**Authors:** Shujie Wang, Dooti Kundu, Xiaohan Zhang, Xinru Tian, Lulu Peng, Kyeong-Man Kim

**Affiliations:** 1Department of Pharmacology, College of Pharmacy, Chonnam National University, Gwang-Ju 61186, Republic of Korea; wangshujie0726@gmail.com (S.W.); dooti.pharm@gmail.com (D.K.); tiandoudou0926@gmail.com (X.T.); lulupeng099@gmail.com (L.P.); 2School of Pharmaceutical Sciences, Guizhou University, Guiyang 550025, China; xhzhang6@gzu.edu.cn

**Keywords:** GRK2, Mdm2, ubiquitination, clathrin, endocytosis, 14-3-3

## Abstract

Upon activation of certain G protein-coupled receptors, Mdm2 promotes the ubiquitination of both GRK2 and arrestin3. Similar to arrestin3, GRK2 ubiquitination was associated with its endocytic activity and proteasomal degradation. Ubiquitination of GRK2 was essential for arrestin3 ubiquitination, and vice versa. Cellular components involved in arrestin3 ubiquitination, including Gβγ, clathrin, and 14-3-3η, were also necessary for GRK2 ubiquitination. Additionally, the arrestin-biased signaling pathway contributed to the ubiquitination of both GRK2 and arrestin3. By employing Mdm2-knockdown cells alongside GRK2 and arrestin3 mutants deficient in ubiquitination sites, as well as receptors lacking phosphorylation sites, we established that the ubiquitinated forms of GRK2 and arrestin3 facilitate clathrin-dependent endocytosis, whereas non-ubiquitinated GRK2 and arrestin3 are responsible for caveolar and a distinct third endocytic pathway, respectively. In the context of clathrin-mediated endocytosis, arrestin3’s interaction with clathrin and GRK2’s interaction with the β2-adaptin subunit of adaptor protein complex 2 were critical. These findings suggest that GRK2 and arrestin3 ubiquitination are mutually dependent, with their ubiquitination states determining their roles in distinct endocytic pathways.

## 1. Introduction

Upon agonist stimulation, G protein-coupled receptors (GPCRs) are phosphorylated by GPCR kinases (GRKs), facilitating arrestin binding and subsequent endocytosis [1]. Receptor internalization occurs primarily via clathrin-mediated endocytosis (CME) or clathrin-independent mechanisms [2].

In CME, adaptor proteins such as arrestins or adaptor protein complex 2 (AP-2) recognize sorting motifs within the cytoplasmic domains of receptors and clathrin [3,4], recruiting clathrin triskelia to form a lattice-like coat [5]. This process is further facilitated by accessory proteins such as epsin and amphiphysin, leading to the formation of clathrin-coated pits. The GTPase dynamin then mediates vesicle scission from the plasma membrane, generating endocytic vesicles [6].

Caveolar endocytosis, a clathrin-independent pathway, is mediated by caveolae—specialized lipid raft domains enriched in cholesterol, sphingolipids, and caveolin proteins [7]. Caveolin-binding motifs within receptor sequences facilitate their partitioning into caveolae. Upon receptor activation, caveolae invaginate, and dynamin-dependent scission results in the release of caveolar vesicles into the cytoplasm, similar to CME [8].

With agonistic stimulation of most G protein-coupled receptors (GPCRs), for example, dopamine D_2_ receptor (D_2_R) and β_2_ adrenergic receptors (β_2_AR), the Gα and Gβγ subunit of the cognate heterotrimeric G protein for the receptor dissociates [9]. The released Gβγ subunit along with the phospholipid of the plasma membrane, binds to the PH (pleckstrin homology) domain of GRK2 (GPCR kinase) located in the carboxyl terminus region, resulting in the recruitment of the inactive cytoplasmic GRK2 toward near the plasma membrane where GPCRs bound to the agonist are located [10,11,12]. This ultimately causes allosteric rearrangement of the AST (active site tether) loop and catalytic domain closure, which are crucial for facilitating its interaction with GPCRs and promoting its catalytic activity towards phosphorylation of the activated receptors [13,14].

In addition to allosteric activations, posttranslational modifications observed in the RH domain or C-terminal region also play important roles in the functional regulation of GRK2. For example, when the αN-helix (Tyr13) or RH (regulator of G protein signaling homology) domain (Tyr86, Tyr92) of GRK2 is phosphorylated by c-Src or EGFR [15,16], the catalytic activity of GRK2 increases. Kinases dependent on second messengers, such as PKA (protein kinase A) and PKC (protein kinase C), also phosphorylate GRK2 and modulate its activity [17,18]. GRK2 also undergoes ubiquitination following agonistic stimulation of some GPCRs.

Ubiquitination is a series of cellular processes where ubiquitin is attached to the lysine residues of target proteins [19]. Among the three key enzymes involved in ubiquitination (E1–E3), the E3 ubiquitin ligase provides substrate selectivity [20]. The ubiquitination of GRK2 and arrestin3, crucial components in GPCR regulation, is mediated by Mdm2 (mouse double minute 2 homology), a member of the RING (really interesting new gene) class E3 ligases [15,21,22,23].

The ubiquitination and degradation of GRK2 and arrestin3 increase with agonistic stimulation of most GPCRs [21,24]. Along with this conventional role of ubiquitination, arrestin3 gains endocytic activity via ubiquitination [21]. In contrast, it is not known whether GRK2 also obtains similar functional roles when ubiquitinated via Mdm2.

Previous studies have linked Mdm2-mediated ubiquitination to the proteasomal degradation of GRK2 [15], and our pilot studies additionally suggested that it also facilitates the endocytic function of GRK2 on receptors. To investigate this further, we analyzed the mechanism of GRK2 ubiquitination, specifically examining the signaling components involved in comparison with the ubiquitination of arrestin3. We also identified the endocytic pathways and the determining factors adopted by the ubiquitination status of GRK2 and arrestin3.

## 2. Results

### 2.1. The Ubiquitination of GRK2 Mediated by Mdm2 Is Associated with Its Endocytic Activity

Unlike the traditional role of ubiquitinating proteins as the proteasomal degradation of the target proteins, Mdm2-mediated ubiquitination of arrestin3 is essential for its endocytic activity. Therefore, we first tested whether the ubiquitination similarly enhances the endocytic activity of GRK2, as it does for arrestin3.

For this, we generated a GRK2 mutant (4KR-GRK2) with key ubiquitination sites (K19, K20, K30, K31) mutated, based on previous studies [23]. Treating cells expressing the dopamine D_2_ receptor (D_2_R) with dopamine (DA) resulted in the ubiquitination of WT-GRK2, but not 4KR-GRK2, indicating that the mutation effectively prevents ubiquitination at the targeted sites (Figure 1A). In GRK2-knockdown (KD) cells expressing D_2_R, receptor endocytosis was more extensively increased with the co-expression of WT-GRK2 compared to 4KR-GRK2 (Figure 1B). These results suggest that GRK2 ubiquitination by Mdm2 is not only involved in the degradation of GRK2, as previously reported [15] but also plays a certain role in the receptor endocytic process.

### 2.2. Arrestin-Biased Signaling Pathway Is Involved in the GRK2 Ubiquitination

It is widely acknowledged that Gβγ subunits, released during GPCR agonist stimulation, and phospholipids from the plasma membrane are crucial for recruiting and activating GRK2 [10,12]. Contrary to this G protein-dependent activation, studies on the D_2_R have shown that GRK2 can be recruited independently of G proteins [25]. Given the requirement of ubiquitination for GRK2 activation, it is likely that GRK2 ubiquitination occurs via an arrestin-biased rather than a G protein-biased signaling pathway.

To elucidate the specific signaling pathways involved in GRK2 ubiquitination, we utilized biased D_2_R variants and biased D_2_R agonists. The biased D_2_R variants, D_2_G (L125N, Y133L) and D_2_Arr (A135R, M140D), are known to preferentially signal through G protein and arrestin pathways, respectively [26,27]. Upon DA treatment, only D_2_Arr mediated GRK2 ubiquitination, not D_2_G (Figure 2A). This was further confirmed using biased D_2_R agonists: DA (balanced), UNC9994 (arrestin-biased) [28], and MLS1547 (G protein-biased) [29]. GRK2 ubiquitination was induced by DA and UNC9994, but not by MLS1547 (Figure 2B).

The involvement of the arrestin-biased signaling pathway in GRK2 ubiquitination suggests that arrestin-mediated processes might influence GRK2 ubiquitination.

### 2.3. Ubiquitination of arrestin3 and GRK2 Is Mutually Related

It has been reported that arrestin is essential for the ubiquitination of GRK2 [15] (Figure 3A). In turn, we aimed to determine whether GRK2 is also necessary for the ubiquitination of arrestin3. As illustrated in Figure 3B, knocking down GRK2 inhibited the ubiquitination of arrestin3, indicating that both proteins are needed for their respective ubiquitination.

Mdm2 is responsible for the ubiquitination of both arrestin3 and GRK2, and recent studies have highlighted that the ubiquitination of arrestin3 is crucial for the ubiquitination of GRK2 [30]. As shown in Figure 3C, the interaction between GRK2 and Mdm2, which was enhanced following D_2_R agonist stimulation, was disrupted when cellular arrestins were knocked down. The co-expression of WT-arrestin3 restored the GRK2-Mdm2 interaction, while K11/12R-arrestin3, which does not bind Mdm2 [31], did not. Conversely, GRK2 knockdown decreased arrestin3 ubiquitination and reduced the interaction between arrestin3 and Mdm2. This effect was reversed by WT-GRK2, but not by 4KR-GRK2, which cannot be ubiquitinated (Figure 3D). These findings indicate that the ubiquitination of GRK2 and arrestin3 is interconnected and that each protein is essential for the other’s ubiquitination.

This interplay between arrestin and GRK2 could be crucial in fine-tuning receptor endocytosis and downstream signaling pathways, highlighting the complex regulatory mechanisms that ensure precise cellular responses to external stimuli.

### 2.4. Cellular Components Responsible for the Ubiquitination of arrestin3 Are Also Involved in the Ubiquitination of GRK2

Since the ubiquitination of arrestin3 is necessary for the ubiquitination of GRK2, the same components responsible for arrestin3 ubiquitination would also be required for the GRK2 ubiquitination. Prior research has shown several crucial factors involved in arrestin3 ubiquitination [22]. These include clathrin heavy chain (CHC), which is essential for forming clathrin-coated pits; Gβγ, which recruits GRK2 to activated GPCRs [10]; GRK-mediated receptor phosphorylation, which facilitates receptor endocytosis; and 14-3-3η, which influences receptor endocytosis, recycling, and downstream signaling [32].

Knockdown of CHC, but not caveolin1 (Cav1), inhibited the D_2_R-mediated GRK2 ubiquitination (Figure 4A). Co-expression of GRK2-CT, which sequesters free Gβγ [33], blocked the D_2_R-mediated GRK2 ubiquitination (Figure 4B). In contrast to WT-β_2_AR, GRK2-KO-β_2_AR, whose consensus sites for GRK2-mediated phosphorylation are mutated [34], failed to mediate the β_2_AR-mediated GRK2 ubiquitination (Figure 4C). Additionally, knockdown of 14-3-3η also prevented the β_2_AR-mediated GRK2 ubiquitination (Figure 4D).

These results suggest that the ubiquitination of GRK2 is intricately linked to the mechanisms governing arrestin3 ubiquitination. Specifically, components such as clathrin heavy chain, Gβγ, GRK-mediated receptor phosphorylation, and 14-3-3η are crucial for facilitating the ubiquitination process of GRK2, highlighting their broader role in receptor signaling and endocytosis.

### 2.5. Differential Endocytic Pathways of D_2_R Are Regulated by the Ubiquitination Status of GRK2 and arrestin3

Our previous findings demonstrated that the CME pathway is associated with ubiquitinated (Ub) forms of arrestin3 and GRK-mediated receptor phosphorylation [35]. Because the ubiquitination of GRK2 is related to the receptor endocytosis (Figure 1) and the ubiquitination of arrestin3 (Figure 3), we investigated whether the ubiquitination status of GRK2 and arrestin3 is related to the different endocytic pathways.

To further investigate the specific endocytic pathways involved in D_2_R endocytosis mediated by the Ub and non-Ub forms of GRK2 and arrestin3, we employed WT-D_2_R and a phosphorylation-deficient mutant of D_2_R (D_2_R-IC23) and conducted experiments in cells with CHC or Cav1 knockdown (CHC-KD and Cav1-KD) (Figure 5A). In D_2_R-IC23, all serine and threonine residues in the second and third intracellular loops of D_2_R were mutated and have been used for the study of phosphorylation-independent processes [36]. As a result, D_2_R-IC23 undergoes endocytosis exclusively through non-CME pathways, including caveolar endocytosis and a third distinct non-clathrin-mediated/non-caveolar endocytic route [35].

In the cells expressing wild-type D_2_R (WT-D_2_R), which supports the ubiquitination of GRK2 and arrestin3, D_2_R endocytosis was inhibited when CHC or Cav1 was knocked down (Figure 5A, compare the three WT-D_2_R groups), suggesting that clathrin-mediated, caveolar, and possibly other pathways are involved in the endocytosis of D_2_R.

To determine the relationship between specific endocytic pathways and the ubiquitination status of GRK2 and arrestin3, we used D_2_R-IC23, a receptor variant incapable of inducing GRK2 and arrestin3 ubiquitination [30]. The endocytosis of WT-D_2_R was more extensive than that of D_2_R-IC23 (Figure 5A, compare the mock groups of WT-D_2_R and D_2_R-IC23). WT-D_2_R endocytosis was inhibited by the knockdown of either CHC or Cav1 (Figure 5A, Mock/WT-D_2_R), whereas D_2_R-IC23 endocytosis was blocked by Cav1 knockdown but remained unaffected by CHC knockdown (Figure 5A, Mock/D_2_R-IC23). These findings align with the framework in Figure 5B, which suggests that WT-D_2_R endocytosis involves both clathrin-mediated and caveolar pathways and is mediated by both ubiquitinated and non-ubiquitinated forms of GRK2 and arrestin3. In contrast, D_2_R-IC23, which undergoes non-clathrin-mediated endocytosis [35], is internalized through non-ubiquitinated GRK2 and arrestin3.

GRK2-mediated enhancement of D_2_R-IC23 endocytosis was unaffected by CHC knockdown but was almost completely abolished in Cav1-KD cells (Figure 5A, D_2_R-IC23, GRK2 groups). Similarly, arrestin3-mediated enhancement of D_2_R-IC23 endocytosis was not impacted by CHC knockdown. However, unlike GRK2, arrestin3’s effect persisted in Cav1-KD cells (Figure 5A, D_2_R-IC23, Arr3 group). The partial inhibition of D_2_R-IC23 endocytosis in Cav1-KD cells is likely due to endogenous non-Ub GRK2 facilitating caveolar endocytosis.

These findings reveal that endocytic pathway selection is governed by the ubiquitination status of GRK2 and arrestin3. Ubiquitinated forms predominantly mediate CME, while non-Ub GRK2 mediates caveolar endocytosis. Distinctively, non-Ub arrestin3 mediates a third endocytic pathway that is non-CME and non-caveolar endocytosis (Figure 5C).

Next, we investigated whether the third endocytic pathway operates in a dynamin-dependent manner. To examine this, we used D_2_R-IC23, GRK2-KD cells, and K44A-dynamin2, a dominant-negative mutant of dynamin2 [2,8]. In GRK2-KD cells expressing D_2_R-IC23, only non-Ub arrestin3 is available, and D_2_R-IC23 undergoes internalization via the third endocytic pathway.

As shown in Figure 5D, pretreatment with MβCD, an inhibitor of caveolar endocytosis, suppressed receptor internalization in Con-KD cells but had no effect in GRK2-KD cells, where only non-Ub arrestin3 is present. Furthermore, the expression of K44A-dynamin2 inhibited receptor endocytosis in both Con-KD and GRK2-KD cells. These findings suggest that similar to clathrin-mediated and caveolar endocytosis, the third endocytic pathway also requires dynamin for its function.

### 2.6. Mdm2-Mediated Ubiquitination Is Involved in the Clathrin-Mediated Endocytosis of Dopamine D_2_ Receptor and β_2_ Adrenoceptor

The conclusion that non-ubiquitinated GRK2 and non-ubiquitinated arrestin3 facilitate caveolar endocytosis and a third distinct endocytic pathway was derived from experiments involving the overexpression of GRK2 and arrestin3. This conclusion was further validated through knockdown studies targeting CHC, Cav1, Mdm2, GRK2, or arrestins. Additionally, we aimed to determine whether the principles established from D_2_R endocytosis could also apply to the endocytosis of β_2_AR.

In the Con-KD cells, CME, caveolar endocytosis, and a third endocytic route are available; both Ub and non-Ub GRK2 and arrestin3 are available (refer to Figure 5B). In the Con-KD cells, the endocytosis of β_2_AR was reduced by about half following treatment with 3 mM MβCD or co-expression of epsin (204–458) that specifically blocks the CME. These observations suggest that both clathrin-mediated and caveolar endocytic pathways, and possibly a third endocytic path are involved in the endocytosis of β_2_AR.

In the CHC-KD cells, caveolar and a third endocytic pathways are available; only non-Ub forms of GRK2 and arrrestin3 are available. As expected, MβCD treatment which blocks caveolar endocytosis inhibited receptor endocytosis, but co-expression of the epsin fragment (204–458), which selectively blocks CME [2], did not affect this inhibition.

In the Cav1-KD cells, CME and a third endocytic pathways are available; both Ub and non-Ub forms of GRK2 and arrestin3 are available. As expected, co-expression of the epsin fragment (204–458) inhibited D_2_R endocytosis, whereas MβCD treatment did not.

Treatment with 10 μM HLI373, an Mdm2 inhibitor, significantly reduced DA-induced endocytosis of the D_2_R. Notably, this inhibitory effect was nullified by the knockdown of CHC, but remained unchanged with the knockdown of Cav1 (Figure 6B). Considering that neither GRK2 nor arrestin3 is ubiquitinated in the absence of CHC, these results indicate that the ubiquitination of GRK2 and/or arrestin3 is essential for the CME. These results were confirmed with Mdm2-KD cells. As shown in Figure 6C, the co-expression of the epsin (204–458) inhibited the D_2_R endocytosis in Con-KD cells, but this inhibition was abolished in Mdm2-KD cells. This result reinforces the idea that Mdm2-mediated ubiquitination is crucial for the CME of GPCRs like β_2_AR or D_2_R.

### 2.7. Interdependent Relationship Between GRK2/arrestin3 Ubiquitination and D_2_R Endocytic Pathways

Next, the relationship between protein ubiquitination status and specific endocytic pathways was confirmed through a series of knockdown experiments focusing on GRK2 and arrestins. As shown in Figure 7A, treatment with the Mdm2 inhibitor HLI373 lowered D_2_R endocytosis, but this effect was abolished upon knockdown of either GRK2 or arrestins, suggesting an interdependent relationship between GRK2 and arrestins in ubiquitination-dependent CME of D_2_R.

Initial characterization revealed that D_2_R endocytosis involves both CME and caveolar pathways in control conditions (Figure 7B, Con-KD). Given that GRK2 and arrestin3 reciprocally regulate their ubiquitination (Figure 3C,D), we performed sequential knockdowns to delineate their specific roles in endocytic pathway selection. In arrestin-KD cells of Figure 7B, where GRK2 ubiquitination is inhibited and only non-Ub GRK2 is available, D_2_R endocytosis exhibited selective sensitivity to MβCD, validating that non-Ub GRK2 facilitates caveolar endocytosis. Conversely, in GRK2-KD cells of Figure 7B, where arrestin3 ubiquitination is blocked, D_2_R endocytosis became resistant to both MDC and MβCD treatments, suggesting that non-Ub arrestin3 mediates neither CME nor caveolar endocytosis but instead facilitates a distinct third endocytic pathway.

To further confirm these outcomes, we conducted rescue experiments using the ubiquitination mutants of GRK2 and arrestin3 (Figure 7C,D). As shown in Figure 7C, the knockdown of GRK2 abolished the sensitivity of D_2_R endocytosis to MβCD and MDC (compare the Mock groups of Con-KD and GRK2-KD cells). The co-expression of WT-GRK2 restored sensitivity to both MβCD and MDC, indicating the availability of all three endocytic routes. However, expression of non-ubiquitinatable 4KR-GRK2 resulted in sensitivity only to MβCD, consistent with the operation of caveolar and third endocytic pathways through non-Ub GRK2 (Figure 7C).

In contrast to GRK2-KD cells (Figure 7C), where sensitivity to both MβCD and MDC was lost, arrestin knockdown specifically abolished sensitivity to MDC (Figure 7D), indicating that non-ubiquitinated GRK2 plays a key role in caveolar endocytosis. Co-expression of WT-arrestin2, which enables the presence of both Ub and non-Ub forms of GRK2 and arrestins, restored sensitivity to both inhibitors. Conversely, co-expression of the non-ubiquitinatable NLSX-arrestin3, which permits only non-Ub forms of GRK2 and arrestin3, restored sensitivity only to MβCD. This relationship between ubiquitination and receptor endocytic pathways confirms that preventing GRK2 ubiquitination redirects endocytosis toward caveolar and a third distinct pathway, mediated by non-ubiquitinated GRK2 and arrestin3, respectively (Figure 7D).

These findings suggest that ubiquitinated GRK2 and arrestin3 drive clathrin-mediated endocytosis (CME), while their non-ubiquitinated forms mediate distinct pathways—non-ubiquitinated GRK2 facilitates caveolar endocytosis, whereas non-ubiquitinated arrestin3 mediates a separate third pathway (Figure 5C). The reciprocal regulation of ubiquitination between GRK2 and arrestin3 adds a layer of complexity to receptor endocytosis regulation, where changes in the cellular levels of either protein can have broad implications for pathway selection.

### 2.8. Ubiquitination-Dependent Protein Interactions of arrestin3 and GRK2 Determine Their Selective Contribution to Distinct Endocytic Pathways

Given that the ubiquitination status of GRK2 and arrestin3 determines the selectivity of the endocytic routes they mediate, we examined the relationship between their ubiquitination status and interaction with clathrin and/or Cav1. For this, we examined the interactions between WT and mutants of arrestin3 and GRK2 versus clathrin and/or caveolin1.

Co-immunoprecipitation experiments revealed that WT-arrestin3, which could exist both in Ub form or non-Ub form, interacted with both CHC and Cav1 (Figure 8A, left three lanes), suggesting its role in both CME and caveolar endocytosis. On the other hand, NLSX-arrestin3, which cannot be ubiquitinated [22], showed no interaction with either protein (fourth and fifth lanes), aligning with its role in the third endocytic pathway, which is non-clathrin and non-caveolar. It is not clear at this point whether Ub-arrestin3 mediates caveolar endocytosis as well.

WT-GRK2 demonstrated its binding to CHC at 2 min and Cav1 at 2 and 5 min (Figure 8B, left four lanes), supporting its function in mediating both endocytic pathways. Unlike non-Ub arrestin3, NLSX-arrestin3 which interacted with neither CHC nor Cav1, 4KR-GRK2, a non-Ub form of GRK2, interacted with both CHC and Cav1 (right three lanes).

These findings indicate that for arrestin3, the choice between CME and non-CME pathways based on ubiquitination status is influenced by its interaction with the CHC. In contrast, the selective involvement of GRK2 in CME based on its ubiquitination status suggests that its interaction with other proteins could play a critical role.

Adaptor protein complex 2 (AP2 adaptor), which is composed of four subunits, plays a central role in the formation of clathrin-coated vesicles [37]. The clathrin binding site is located on the β2 subunit (β2 adaptin) [3], and the interaction between β2 adaptin and GRK2 or arrestin3 was determined. Both the WT- and the non-Ub mutant of arrestin3 (K11/12R) interacted with β2 adaptin (Figure 8C). However, only the WT form of GRK2, and not the non-Ub mutant (4KR-GRK2), interacted with β2 adaptin (Figure 8D). These results highlight distinct mechanisms underlying CME selectivity: arrestin3’s ubiquitination directly governs its interaction with clathrin heavy chain (CHC), while GRK2’s ubiquitination regulates CME via its binding to β2 adaptin.

## 3. Discussion

This study underscores the multifaceted roles of ubiquitination in the regulation and functionality of GRK2, particularly in its involvement in endocytic pathways of GPCRs. Our findings extend the known regulatory framework governing GRK2 activity and highlight the complex interplay between GRK2 and arrestin3 ubiquitination processes, suggesting a coordinated mechanism that ensures precise cellular responses.

In agreement with traditional aspects of ubiquitination, it has been reported that the ubiquitination of GRK2 plays a role in facilitating its degradation [23]. However, our study provides compelling evidence that Mdm2-mediated ubiquitination of GRK2 extends beyond this conventional role, endocytic activity. This parallels the established role of ubiquitination in enhancing the endocytic activity of arrestin3 [21], suggesting a similar mechanism for GRK2.

Our results underscore a reciprocal relationship between the ubiquitination of GRK2 and arrestin3. Knockdown of GRK2 or removal of GRK2 ubiquitination prevented the arrestin3 ubiquitination, and vice versa, highlighting a mutual dependency. We also identified shared cellular components governing both GRK2 and arrestin3 ubiquitination, including CHC, Gβγ, receptor phosphorylation, and 14-3-3η [30]. The interdependence between GRK2 and arrestin3 ubiquitination and the shared dependency on common cellular machinery underscores the coordinated regulation of GRK2 and arrestin3 ubiquitination. Notably, GRK2 ubiquitination occurs through arrestin-biased rather than G protein-biased signaling, revealing an unexpected layer of regulation in GPCR trafficking.

Our findings reveal complex regulatory mechanisms governing GPCR endocytosis through the interplay of GRK2 and arrestin3 ubiquitination. The data demonstrates three distinct endocytic pathways: clathrin-mediated endocytosis (CME), caveolar endocytosis, and a novel third pathway, each regulated by the ubiquitination status of GRK2 and arrestin3 (Figure 9). Ubiquitinated GRK2 and arrestin3 primarily mediate CME, while their non-Ub forms direct trafficking through alternative pathways—caveolar endocytosis for GRK2 and the third pathway for arrestin3. This pathway selection is further regulated through distinct protein interactions. arrestin3’s ubiquitination directly controls its binding to CHC and Cav1, while GRK2’s ubiquitination modulates β2 adaptin interaction, suggesting different molecular mechanisms for CME selection.

Identifying a third endocytic pathway, distinct from CME and caveolar endocytosis, adds another layer of complexity to GPCR trafficking regulation. This pathway, mediated by non-Ub arrestin3, provides cells additional flexibility in receptor internalization. The ability to reroute receptor trafficking through alternative pathways when one route is blocked demonstrates the adaptability of the cellular endocytic machinery.

These findings have important implications for cellular signaling regulation. Changes in GRK2 and arrestin3 expression levels or ubiquitination status could significantly alter GPCR trafficking patterns, potentially affecting downstream signaling outcomes. This complex interplay provides cells with precise control over receptor trafficking and may contribute to the diversity of cellular responses to GPCR activation.

Understanding these regulatory mechanisms could have therapeutic implications, particularly for conditions involving dysregulated GPCR signaling. GRK2 is implicated in the regulation of β-adrenergic receptors in the heart. Its overexpression has been associated with heart failure [38], suggesting that targeting GRK2 could be a therapeutic strategy. MDM2’s Influence on GRK2 Activity: MDM2 has been shown to regulate cardiac contractility by inhibiting GRK2-mediated desensitization of β-adrenergic receptors [39], highlighting its potential as a therapeutic target in heart diseases.

The endocytosis of β_2_AR serves as a potential pathway for receptor resensitization [40]. Phosphorylation of D_2_Rs is critical for their recycling and resensitization, and disruption of this process leads to slower receptor recycling and enhanced desensitization [36]. The critical importance of endocytosis in maintaining cellular homeostasis and its involvement in various diseases when dysregulated [41]. The ability to selectively modulate specific endocytic pathways through manipulation of GRK2 and arrestin3 ubiquitination might offer new therapeutic strategies for such disorders.

## 4. Materials and Methods

### 4.1. Materials

Dopamine, isoproterenol, methyl-β-cyclodextrin (MβCD), monodansylcadaverine (MDC), UNC9995, MLS1547, agarose beads coated with monoclonal antibodies against FLAG epitope, rabbit anti-FLAG M2 antibodies (AB_439687), HLI373, an Mdm2 inhibitor, rabbit anti-actin antibody (AB_476694), rabbit 14-3-3η antibody (AB_10747080), and goat anti-mouse (AB_390192) and anti-rabbit (AB_390191) horseradish peroxidase (HRP)-labeled secondary antibodies were sourced from Sigma-Aldrich Chemical Co. (St. Louis, MO, USA). Mouse monoclonal antibodies targeting the clathrin heavy chain (CHC) (AB_397865) and caveolin1 (Cav1) (AB_397472) were obtained from BD Biosciences (San Jose, CA, USA). Monoclonal antibodies against the HA epitope (AB_783679), GRK2 (AB_626751), and Mdm2 (AB_627920) were procured from Santa Cruz Biotechnology (Santa Cruz, CA, USA). Antibodies to β-actin (AB_10950489), lamin b1 (AB_10896336), and rabbit 14-3-3η monoclonal antibody (AB_10829034) were obtained from Cell Signaling Technology (Danvers, Massachusetts, USA). [^3^H]-Sulpiride (84 Ci/mmol) and [^3^H]-CGP12177 were obtained from PerkinElmer Life Sciences (Boston, MA, USA).

### 4.2. DNA Constructs

Dopamine D_2_ receptor (D_2_R), FLAG-D_2_R, and D_2_R-IC23 were reported previously [36,42]. In D_2_R-IC23, all serine and threonine residues located within the 2nd and 3rd intracellular loops were mutated to alanine and valine residues. Two D_2_R mutants that preferentially signal through G protein (D_2_G, L125N/Y133L) or arrestin (D_2_Arr, A135R/M140D) were previously reported [27]. A mutant of GRK2 deficit of Mdm2-mediated ubiquitination sites (4KR-GRK2, L19, 20, 30, 31R) [23] was prepared by site-directed mutagenesis. FLAG-D_2_Arr and FLAG-D_2_G were prepared by polymerase cycle reaction. GRK2-CT and FLAG-β2 adaptin were described previously [43,44]. β_2_ Adrenergic receptor, GRK2, FLAG-GRK2, arrestin3, FLAG-arrestin3, K11/12R-arrestin3, NLSX-arrestin3, arrestin3-GFP, FLAG-Mdm2, and HA-Ub were described previously [2,26,42].

### 4.3. Cell Culture

Human embryonic kidney (HEK-293) cells were sourced from the American Type Culture Collection (Manassas, VA, USA) and cultured in minimal essential medium supplemented with 10% fetal bovine serum, 100 units/mL penicillin, and 100 μg/mL streptomycin in a humidified atmosphere with 5% CO_2_. Mdm2-knockdown (KD) cells, GRK2-KD cells, CHC-KD cells, Cav1-KD cells, and 14-3-3η-KD cells were maintained following previously established protocols [2,45,46,47]. Arrestin2-KD and arrestin3-KD cells were established through stable transfection of the shRNA of arrestin2 in pLKO.1 (Santa Cruz Biotechnology) and the shRNA in pcDNA3.0 [36]. Arrestin2/3-KD cells were established by stable transfection of shRNA of arrestin3 into arrestin2-KD cells.

### 4.4. Immunoprecipitation and Immunoblotting

Cells were rinsed with phosphate-buffered saline (PBS) and lysed using RIPA buffer (150 mM NaCl, 50 mM Tris pH 8.0, 1% NP-40, 0.5% deoxycholate, and 0.1% sodium dodecyl sulfate (SDS)) for 1 h at 4 °C with gentle agitation. The cell lysates were then centrifuged for 30 min at 14,000× *g*, and the supernatants were combined with 35 μL of agarose beads coated with antibodies against FLAG (50% slurry) for 2–3 h at 4 °C with gentle agitation. Subsequently, the beads were washed three times with ice-cold washing buffer (50 mM Tris pH 7.4, 137 mM NaCl, 10% glycerol, and 1% NP-40) for 5 min each, and sodium dodecyl sulfate (SDS) buffer was added. The immunoprecipitates were subjected to SDS polyacrylamide gel electrophoresis (PAGE) using a 5% stacking gel and a 10% separation gel, followed by transfer to nitrocellulose membranes (GE Healthcare Life Sciences; Chicago, IL, USA). The membranes were blocked by incubating with 5% non-fat dry milk in TBS-T (Tris-buffered saline with Tween™ 20 (Junsei Chemical, Tokyo, Japan)) at 20 °C for 2–5 h. They were then washed with TBS-T for 15 min at 20 °C. Next, the membranes were incubated with primary antibodies dissolved in 2% BSA in TBS-T (1:2000 dilution) for 2 h at 20 °C or overnight at 4 °C. After another 15-min wash with TBS-T, the membranes were incubated with the corresponding HRP-conjugated secondary antibodies dissolved in 2% BSA (1:3000 dilution) for 1 h at 20 °C. Protein bands were visualized and quantified using the ChemiDoc MP imaging system (BioRad, Hercules, CA, USA).

### 4.5. Detection of Protein Ubiquitination

HEK-293 cells expressing D_2_R were transfected with HA-ubiquitin (HA-Ub) along with FLAG-tagged GRK2 or arrestin3. After 4–6 h of serum starvation, the cells were treated with 10 μM DA for 2 min. Cell lysates were prepared using lysis buffer containing 150 mM NaCl, 50 mM Tris pH 7.4, 1 mM EDTA, 1% Triton X-100, 10% (*v*/*v*) glycerol, 1 mM sodium orthovanadate, 1 mM sodium fluoride, 2 mM phenylmethylsulfonyl fluoride, 5 μg/mL leupeptin, 5 μg/mL aprotinin, and 10 mM N-ethylmaleimide. Immunoprecipitation was performed using FLAG beads, and the immunoprecipitates were subsequently analyzed by SDS-PAGE and probed with antibodies against HA and FLAG.

### 4.6. Receptor Endocytosis

Endocytosis of D_2_R and β_2_AR was assessed using the hydrophilic properties of [^3^H]-sulpiride and [^3^H]-CGP12177, respectively, as previously described [2,42]. HEK-293 cells expressing D_2_R or β_2_AR were seeded at a density of 1.5 × 10^5^ cells per well in 24-well plates one day after transfection. The following day, cells were rinsed once and pre-incubated for 15 min with 0.5 mL pre-warmed serum-free medium containing 10 mM HEPES (pH 7.4) at 37 °C. Subsequently, the cells were stimulated with 10 μM DA or ISO for 1 h and 20 min, respectively. Afterward, cells were incubated with 250 μL of [^3^H]-sulpiride (2.2 nM) or [^3^H]-CGP12177 (10 nM) at 4 °C for 150 min in the presence or absence of an unlabeled competitive inhibitor (10 μM haloperidol or propranolol). Following incubation, the cells were washed three times with the same medium, and 1% SDS was added. Samples were mixed with 2 mL scintillation fluid and counted using a liquid scintillation analyzer (Perkin Elmer, 1450 MicroBeta TriLux, Waltham, MA, USA).

### 4.7. Statistics

To minimize variation between experiments, the normalized values of immunoblots were averaged and expressed as fold change, with the mean value of the control group set as 1. The *Y*-axis label in the figures represents the “fold mean of the controls”. The analysis used GraphPad Prism 8 software (GraphPad Software Inc., San Diego, CA, USA). Data were expressed as the mean ± SD. Statistical significance was assessed using a paired two-tailed Student’s *t*-test for two groups or one-way ANOVA with Tukey’s post hoc test for multiple groups. The post hoc tests were conducted only if the F value in ANOVA achieved a significance level of *p* < 0.05 and there was no significant variance inhomogeneity.

## 5. Conclusions

This study reveals the multifaceted roles of Mdm2-mediated ubiquitination of GRK2, demonstrating that it not only facilitates the degradation of GRK2 but also plays a critical role in regulating its endocytic activity. Importantly, the ubiquitination of GRK2 and arrestin3 is shown to involve shared molecular factors, highlighting an interdependent relationship between these two proteins. Moreover, the findings suggest that the arrestin-biased signaling pathway is involved in the ubiquitination of GRK2, adding a new layer of complexity to the regulation of these processes. The study further underscores the role of ubiquitination status in dictating the specific endocytic pathways mediated by GRK2 and arrestin3 based on their ubiquitination status. Ubiquitination-dependent selective interactions with key adaptor proteins, such as clathrin heavy chain or β2 adaptin, are shown to be pivotal in determining the choice of endocytic routes. Collectively, these results provide critical insights into the intricate regulatory mechanisms of GPCRs, emphasizing the sophisticated interplay between ubiquitination, endocytosis, and signaling.

## Figures and Tables

**Figure 1 ijms-26-03238-f001:**
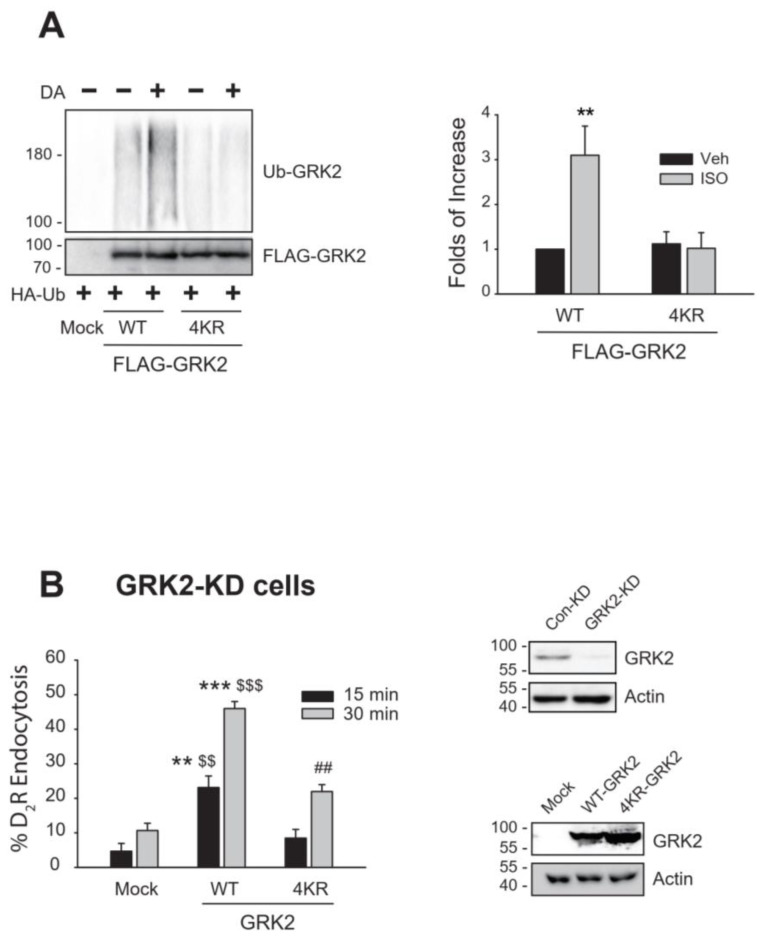
Association between Mdm2-mediated ubiquitination of GRK2 and its endocytic activity. (**A**) HEK-293 cells expressing D_2_R (between 1.5 and 1.7 pmol/mg protein) were transfected with HA-Ub together with either FLAG-tagged WT- or 4KR-GRK2. Cells were treated with 10 μM DA for 2 min. Cell lysates were immunoprecipitated with FLAG beads and the immunoprecipitates were blotted with antibodies against HA and FLAG to detect the Ub-GRK2 and total GRK2, respectively. ** *p* < 0.01 compared to other groups (n = 3). (**B**) GRK2-knockdown (KD) HEK-293 cells were transfected with D_2_R along with a mock vector, WT-GRK2, or 4KR-GRK2. Cells were treated with 10 μM DA for 1 h, followed by three washes with serum-free media. Cells were treated with 2.2. nM [^3^H]-sulpiride at 150 min at 4 °C. ** *p* < 0.01, *** *p* < 0.001 compared to 15 min/Mock group; ^$$^
*p* < 0.01, ^$$$^
*p* < 0.001 compared to 30 min/Mock group; ^##^
*p* < 0.001 compared to 30 min/Mock group (n = 3). Cell lysates from Con-KD and GRK2-KD cells were immunoblotted with antibodies against GRK2 and actin. The knockout efficiency was about 95%. In addition, the cell lysates obtained from the GRK2-KD cells transfected with a mock vector, WT-GRK2, or 4KR-GRK2 were immunoblotted with antibodies against GRK2 and actin.

**Figure 2 ijms-26-03238-f002:**
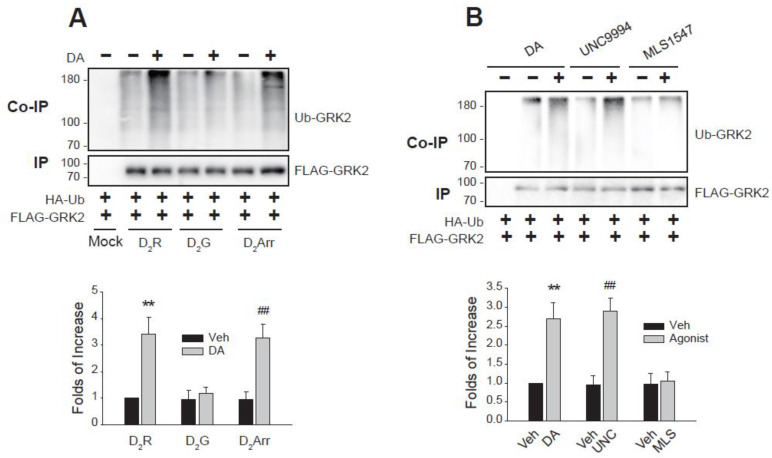
Involvement of arrestin-biased pathway in the Mdm2-mediated GRK2 ubiquitination. (**A**) HEK-293 cells were transfected with HA-Ub and FLAG-GRK2 together with a mock vector, WT-D_2_R, D_2_G, and D_2_Arr. Cells were treated with 10 μM DA for 2 min. ** *p*, ^##^
*p* < 0.01 compared to other groups except for the DA/D_2_Arr group and DA/D_2_R group, respectively (n = 3). (**B**) HEK-293 cells were transfected with HA-Ub, FLAG-GRK2, and D_2_R. Cells were treated with 10 μM DA, UNC9994, or MLS1547 for 2 min. ** *p*, ^##^
*p* < 0.01 compared to other groups except for the UNC9994 group and DA group, respectively (n = 3).

**Figure 3 ijms-26-03238-f003:**
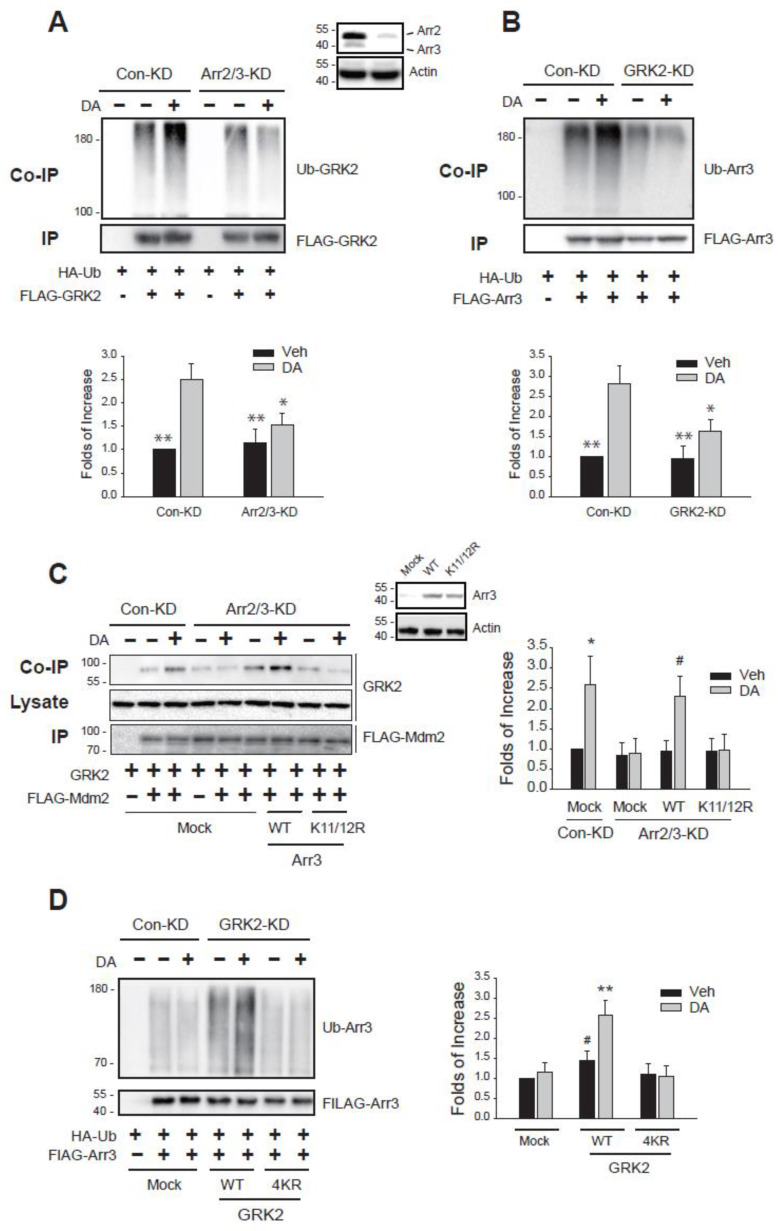
Ubiquitination of arrestin3 and GRK2 is reciprocally related. (**A**) Con-KD and arrestin2/3-KD cells were transfected with D_2_R (1.5–1.7 pmol/mg protein), HA-Ub, and FLAG-GRK2. Cells were treated with 10 μM DA for 2 min. * *p* < 0.05, ** *p* < 0.01 compared to DA/Con-KD group (n = 3). (**B**) Con-KD and GRK2-KD cells were transfected with D_2_R (1.7–2.0 pmol/mg protein), HA-Ub, and FLAG-arrestin3. Cells were treated with 10 μM DA for 2 min. * *p* < 0.05, ** *p* < 0.01 compared to DA/Con-KD group (n = 3). (**C**) Con-KD and arrestin2/3-KD cells were transfected with D_2_R (1.5–1.7 pmol/mg protein), GRK2, FLAG-Mdm2, together with a mock vector, WT-arrestin3, or K11/12R-arrestin3. Cells were treated with 10 μM DA for 2 min. Cell lysates were immunoprecipitated with FLAG beads. Co-IP/lysates and IP were immunoblotted with antibodies against GRK2 and FLAG, respectively. * *p*, ^#^
*p* < 0.05 compared to other groups except the DA/WT/Arr2/3-KD group and DA/Mock/Con-KD group, respectively (n = 3). (**D**) Con-KD and GRK2-KD cells were transfected with D_2_R (1.7–2.0 pmol/mg protein), HA-Ub, FLAG-arrestin3, together with a mock vector, WT-GRK2, or 4KR-GRK2. Cells were treated with 10 μM DA for 2 min. ** *p* < 0.01 compared to other groups except Veh/WT group; ^#^
*p* < 0.05 compared to DA/WT group (n = 3).

**Figure 4 ijms-26-03238-f004:**
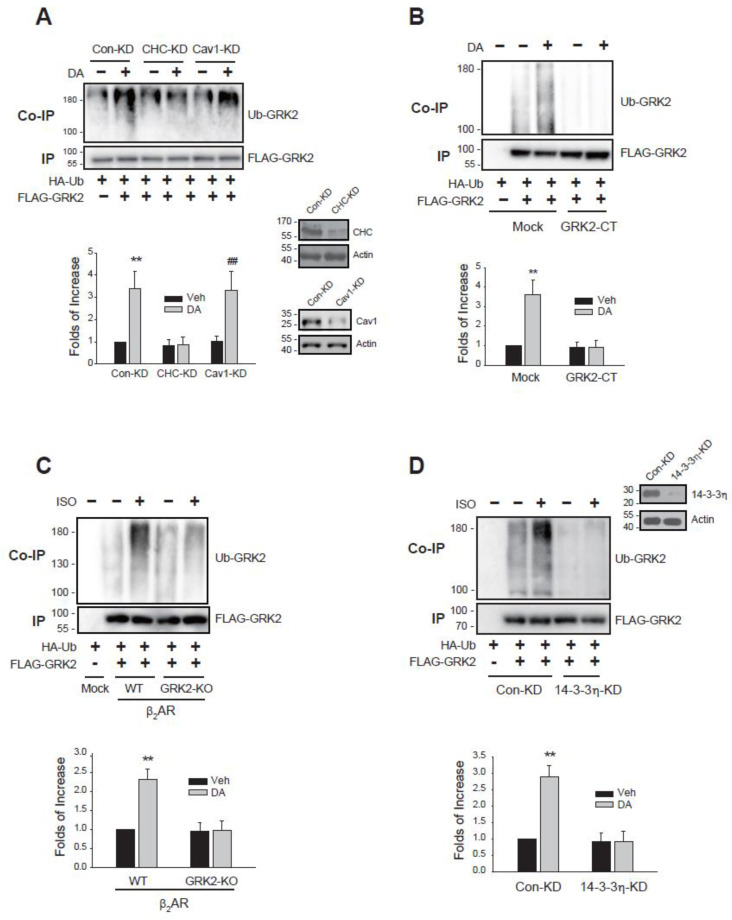
Cellular components responsible for the ubiquitination of GRK2. (**A**) Con-KD, CHC-KD, Cav1-KD were transfected with D_2_R (1.5–1.9 pmol/mg protein), HA-Ub, and FLAG-GRK2. Cells were treated with 10 μM DA for 2 min. ** *p*, ^##^
*p* < 0.01 compared to other groups except for the DA/Cav1-KD group and DA/Con-KD group, respectively (n = 3). Lysates from Con-KD, CHC-KD, and Cav1-KD cells were immunoblotted with antibodies against CHC/actin and Cav1/actin. The knockdown efficiency of CHC-KD and Cav1-KD cells was about 90–95%. (**B**) Cells were transfected with D_2_R (1.7–1.9 pmol/mg protein) together with a mock vector or GRK2-CT. Cells were treated with 10 μM DA for 2 min. ** *p* < 0.01 compared to other groups (n = 3). (**C**) Cells were transfected with HA-Ub and FLAG-GRK2 together with a mock vector, WT-β_2_AR, or GRK2-KO-β_2_AR (1.7–1.9 pmol/mg protein). Cells were treated with 10 μM isoproterenol (ISO) for 2 min. ** *p* < 0.01 compared to other groups (n = 3). (**D**) Con-KD or 14-3-3η-KD cells were transfected with D_2_R (1.7–1.9 pmol/mg protein), HA-Ub, and FLAG-GRK2. Cells were treated with 10 μM DA for 2 min. ** *p* < 0.01 compared to other groups (n = 3). Lysates of Con-KD and 14-3-3η-KD cells were blotted with antibodies against 14-3-3η and actin. The knockdown efficiency of 14-3-3η was about 90%.

**Figure 5 ijms-26-03238-f005:**
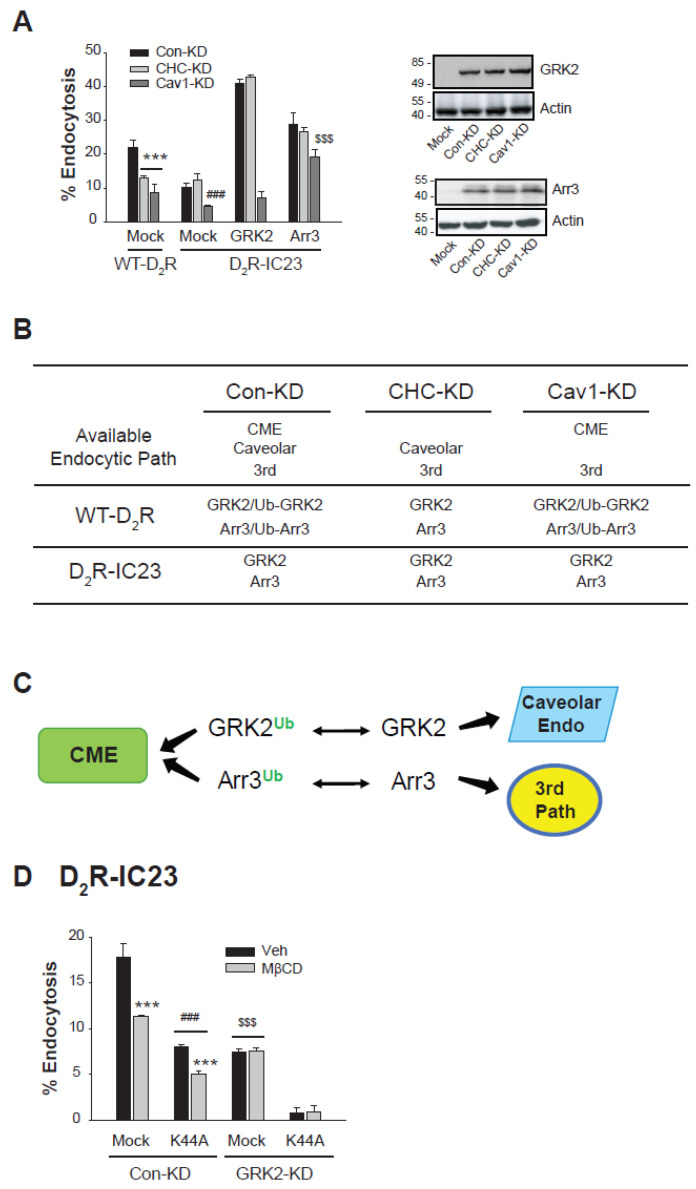
Deubiquitinated GRK2 mediates caveolar endocytosis, whereas deubiquitinated arrestin3 facilitates a distinct endocytic pathway independent of clathrin-mediated and caveolar endocytosis. (**A**) Con-KD, CHC-KD, and Cav1-KD cells were transfected with either WT-D_2_R and a mock vector or D_2_R-IC23 along with a mock vector, GRK2, or arrestin3. Cells were treated with 10 µM DA for 1 h. After three washes, 2.2 nM [^3^H]-sulpiride was applied for 150 min at 4 °C. Statistical significance is indicated as follows: *** *p* < 0.001 compared to the corresponding Con-KD group; ^###^
*p* < 0.001 compared to the corresponding Con-KD and CHC-KD groups; ^$$$^
*p* < 0.001 compared to the Con-KD cells transfected with GRK2 and Arr3 in the D_2_R-IC23-expressing condition (n = 3). Cell lysates from each transfection group were analyzed via Western blot using antibodies against GRK2, arrestin, and actin. (**B**) The ubiquitination status and the available endocytic pathways of D_2_R in different phosphorylation states of D_2_R and expression levels of CHC and Cav1. (**C**) A schematic representation depicting how the ubiquitination status of GRK2 and arrestin3 regulates the selection of specific endocytic pathways. (**D**) Con-KD and GRK2-KD HEK-293 cells were transfected with either a mock vector or K44A-dynamin2 alongside D_2_R-IC23. Cells were treated either with vehicle or 3 mM MβCD for 30 min. Following this, cells were stimulated with 10 μM DA for 2 and 30 min, washed three times, and subsequently incubated with 2.2 nM [^3^H]-sulpiride for 150 min at 4 °C. *** *p* < 0.001 compared to the each Veh treatment group; ^###^
*p* < 0.001 compared to the Mock/Con-KD group; ^$$$^
*p* < 0.001 compared to Mock/Con-KD and K44A/GRK2-KD groups (n = 3).

**Figure 6 ijms-26-03238-f006:**
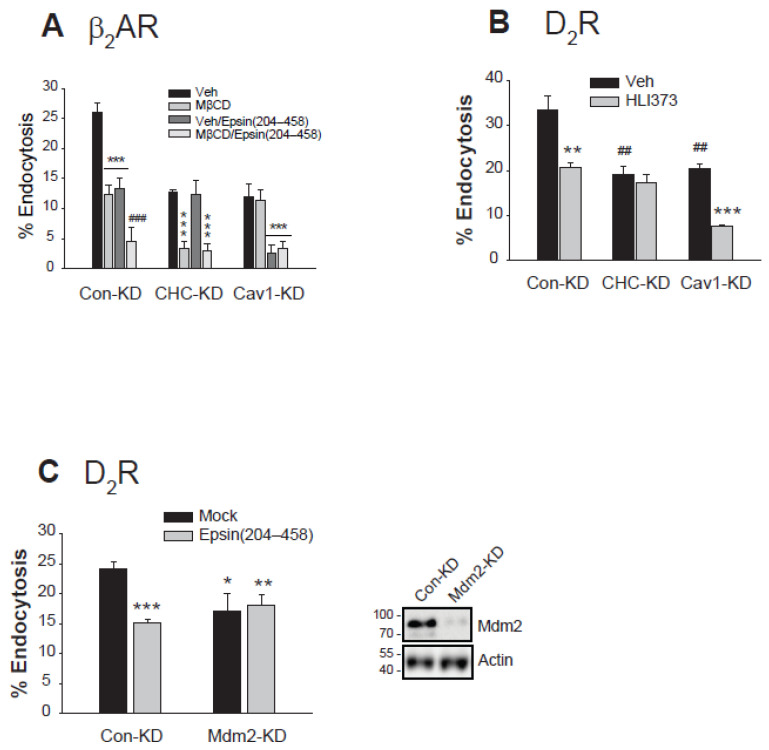
Mdm2-mediated ubiquitination is needed for the clathrin-mediated endocytosis of dopamine D_2_ receptor and β_2_ adrenoceptor. (**A**) Con-KD, CHC-KD, or Cav1-KD HEK-293 cells were transfected with β_2_AR (1.6–1.9 pmol/mg protein) along with either a mock vector or epsin (204–458). The cells were pretreated with either a vehicle or 3 mM MβCD for 30 min, followed by exposure to 10 µM ISO for 20 min. After three washes with serum-free media, the cells were treated with 10 nM [^3^H]-CGP12177 for 150 min at 4 °C. *** *p* < 0.001 compared to each Veh-treated group; ^###^
*p* < 0.001 compared to other groups of Con-KD cells (n = 3). (**B**) Con-KD, CHC-KD, or Cav1-KD HEK-293 cells were transfected with D_2_R (1.6–1.9 pmol/mg protein). Cells were treated with a Veh or HLI373 for 30 min, followed by treatment with 10 μM DA for 1 h. After three washes with a serum-free media, cells were treated with 2.2 nM [^3^H]-sulpiride. ** *p* < 0.01, *** *p* < 0.001 compared to the corresponding Veh cells; ^##^
*p* < 0.01 compared to Veh/Con-KD cells. (**C**) Con-KD and Mdm2-KD HEK-293 cells were transfected with D_2_R (1.7–1.9 pmol/mg protein) along with a mock vector or epsin (204–458). Cells were treated with a vehicle or 10 μM DA for 1 h. After three washes with a serum-free media, cells were treated with 2.2 nM [^3^H]-sulpiride. * *p* < 0.05, ** *p* < 0.01, *** *p* < 0.001 compared to Mock/Con-KD cells (n = 3).

**Figure 7 ijms-26-03238-f007:**
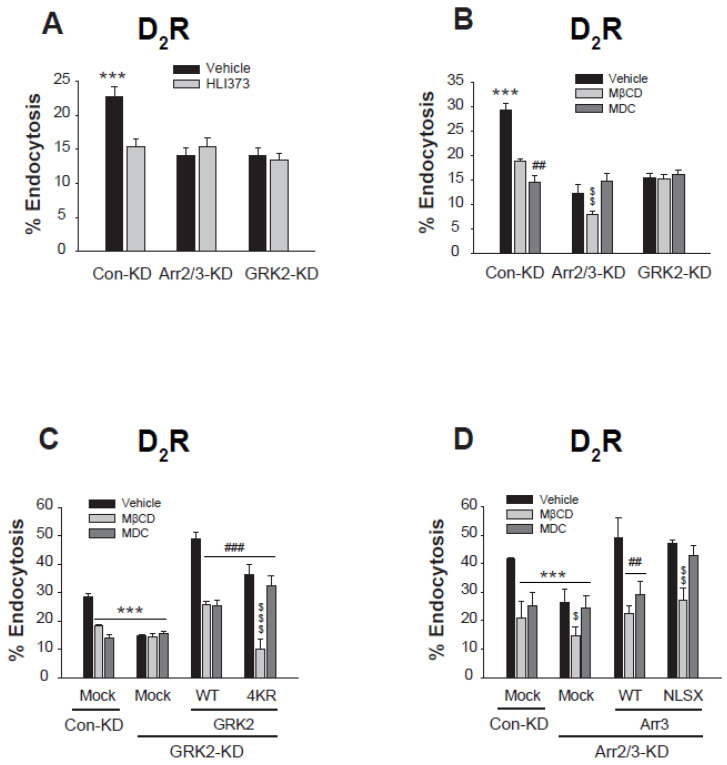
The ubiquitination of GRK2 and arrestin3 is interdependently linked to the clathrin-mediated endocytosis of the dopamine D_2_ receptor. (**A**) Con-KD, arrestin2/3-KD, and GRK2-KD HEK-293 cells were transfected with D_2_R (1.5–1.7 pmol/mg protein). Cells were pretreated with a vehicle or 3 μM HLI373 for 15 h, followed by 10 μM DA for 1 h. After three washes with serum-free media, cells were treated with 2.2. nM [^3^H]-sulpiride for 150 min at 4 °C. *** *p* < 0.001 compared to other groups (n = 3). (**B**) Con-KD, arrestin2/3-KD, and GRK2-KD HEK-293 cells were transfected with D_2_R (1.5–1.7 pmol/mg protein). Cells were pretreated with a vehicle, 3 mM MβCD for 30 min, or 200 μM MDC for 20 min, followed by 10 μM DA for 1 h. *** *p* < 0.001 compared to other groups; ^##^
*p* < 0.01 compared to MβCD/Con-KD cells; ^$$^
*p* < 0.01 compared to the arrestin2/3-KD cells treated with vehicle or MDC (n = 3). (**C**) Con-KD and GRK2-KD HEK-293 cells were transfected with D_2_R (1.5–1.7 pmol/mg protein) along with a mock vector, WT- or 4KR-GRK2. Cells were pretreated with a vehicle, 3 mM MβCD for 30 min, or 200 μM MDC for 20 min, followed by treatment with 10 μM DA for 1 h. *** *p* < 0.001 compared to the Con-KD cells treated with a vehicle; ^###^
*p* < 0.001 compared to the cells transfected with WT-GRK2 and treated with vehicle; ^$$$^
*p* < 0.001 compared to other 4KR-GRK2 groups (n = 3). (**D**) Con-KD and arrestin2/3-KD HEK-293 cells were transfected with D_2_R (1.5–1.7 pmol/mg protein) along with a mock vector, WT- or NLSX-arrestin3. Cells were pretreated with a vehicle, 3 mM MβCD for 30 min, or 200 μM MDC for 20 min, followed by treatment with 10 μM DA for 1 h. *** *p* < 0.001 compared to vehicle-treated group of Con-KD cells; ^##^
*p* < 0.01 compared to the vehicle-treated cells expressing WT-arrestin3; ^$^
*p* < 0.05, ^$$^
*p* < 0.01 compared to corresponding vehicle- and MDC-treated cells (n = 3).

**Figure 8 ijms-26-03238-f008:**
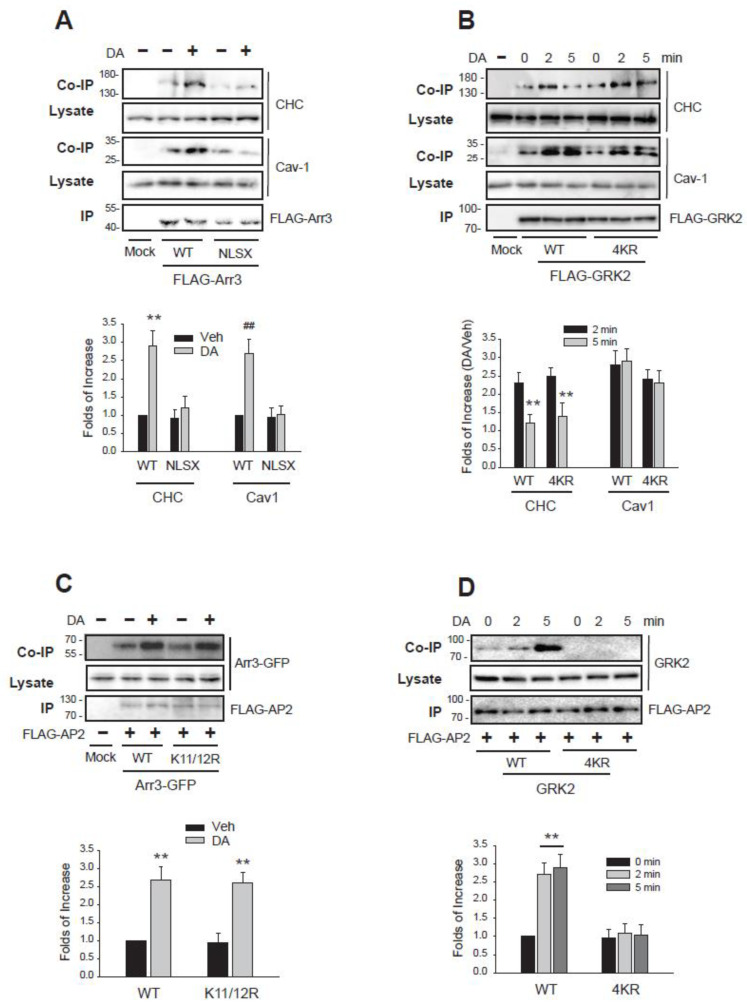
Ubiquitination-mediated interactions between arrestin3 and GRK2 govern their selective roles in different endocytic pathways. (**A**) HEK-293 cells were transfected with D_2_R along with FLAG-tagged WT-arrestin3 or NLSX-arrestin3. Cells were treated with 10 μM DA for 2 min. Lysates were immunoprecipitated with FLAG beads. Co-IP/lysates and IP were immunoblotted with antibodies against endogenous CHC/Cav1 and FLAG, respectively. ** *p* < 0.01, ^##^
*p* < 0.01 compared to corresponding vehicle-treated groups (n = 3). (**B**) HEK-293 cells were transfected with D_2_R along with FLAG-tagged WT-GRK2 or 4KR-GRK2. Cells were treated with 10 μM DA for 2 min. Lysates were immunoprecipitated with FLAG beads. Co-IP/lysates and IP were immunoblotted with antibodies against endogenous CHC/Cav1 and FLAG, respectively. ** *p* < 0.01 compared to corresponding 2 min groups (n = 3). (**C**) HEK-293 cells were transfected with D_2_R and FLAG-tagged β2 adaptin (β2-ad) along with a mock vector, GFP-tagged WT-arrestin3 or K11/12R-arrestin3. Cells were treated with 10 μM DA for 2 min. Lysates were immunoprecipitated with FLAG beads. Co-IP/lysates and IP were blotted with antibodies against GFP and FLAG, respectively. ** *p* < 0.01 compared to corresponding Veh groups (n = 3). (**D**) HEK-293 cells were transfected with D_2_R and FLAG-β2-ad along with a mock vector, WT-GRK2, or 4KR-GRK2. Cells were treated with 10 μM DA for 2 min. Lysates were immunoprecipitated with FLAG beads. Co-IP/lysates and IP were blotted with antibodies against GRK2 and FLAG, respectively. ** *p* < 0.01 compared to other groups (n = 3).

**Figure 9 ijms-26-03238-f009:**
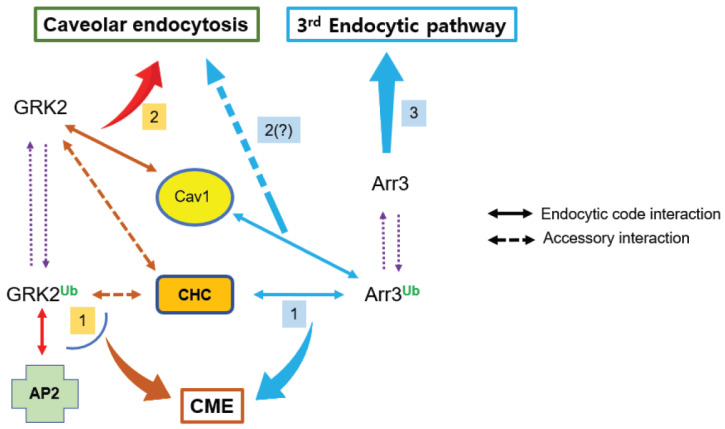
Summary of the relationship between Mmd2-mediated ubiquitination of GRK2/arrestin3 and endocytic pathways. Both GRK2 and arrestin3, when ubiquitinated, regulate clathrin-mediated endocytosis (CME). Distinct protein interactions determine the pathway specificity: Ub-arrestin3 binds to clathrin heavy chain (CHC), while Ub-GRK2 interacts with adaptor protein 2 (AP2). When non-ubiquitinated, these proteins control different endocytic pathways—GRK2 mediates caveolar endocytosis through its interaction with Cav1, whereas arrestin3, which lacks Cav1 binding capability, regulates a distinct third endocytic pathway that remains to be characterized (question mark). Protein interactions depicted by arrows connected with solid lines dictate the endocytic pathway (endocytic code interaction), while arrows linked by dotted lines represent protein interactions that have a supporting role (accessory interaction).

## Data Availability

The data that support the findings of this study are available on request from the corresponding author.

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
