# Peer review of "Mdm2-Mediated Ubiquitination Plays a Pivotal Role in Differentiating the Endocytic Roles of GRK2 and Arrestin3"

_ijms, 2025, doi:10.3390/ijms26073238_

Round 1
Reviewer 1 Report
Comments and Suggestions for Authors
As a reviewer, I would like to point out that this study provides valuable information on the function of Mdm2-mediated ubiquitination in the regulation of GRK2 and arrestin3 endocytic processes. The article demonstrates that Mdm2-mediated ubiquitination of GRK2 is involved in proteasomal degradation and receptor endocytosis. The article also demonstrates that the ubiquitination status of GRK2 and arrestin3 dictates their engagement in different endocytic pathways. However, I do believe that the following remarks should be addressed before publication.
1. This research study is not experimentally confirmed enough to support its results. The employment of overexpression and knockdown experiments may not reflect the physiological relevance of the observed effects. Incorporation of experiments with endogenous proteins or more clinically relevant models would increase the strength of the findings and offer a greater understanding of the mechanisms at play.
2. This study does not mention the potential therapeutic implications or clinical relevance of the findings. The authors must, apart from that, speak to how their findings could be applied in models of disease or drug discovery, which will raise the general importance and relevance of the study.
3. I did find some similarities between the manuscript and previously published work. I would recommend that the authors correct these concerns to avoid any ethical issues related to plagiarism. However, I do believe that the following remarks must be addressed before publication.
In conclusion, although the study presents fascinating facts about regulating GRK2 and arrestin3 by Mdm2-mediated ubiquitination, the lack of robust experimental evidence and inadequate discussion of clinical relevance hinder its recommendation for publication in this highly regarded journal. The authors should address these concerns and provide additional experimental evidence before resubmission of the manuscript.
Author Response
Comments from Reviewer-1
As a reviewer, I would like to point out that this study provides valuable information on the function of Mdm2-mediated ubiquitination in the regulation of GRK2 and arrestin3 endocytic processes. The article demonstrates that Mdm2-mediated ubiquitination of GRK2 is involved in proteasomal degradation and receptor endocytosis. The article also demonstrates that the ubiquitination status of GRK2 and arrestin3 dictates their engagement in different endocytic pathways. However, I do believe that the following remarks should be addressed before publication.
- This research study is not experimentally confirmed enough to support its results. The employment of overexpression and knockdown experiments may not reflect the physiological relevance of the observed effects. Incorporation of experiments with endogenous proteins or more clinically relevant models would increase the strength of the findings and offer a greater understanding of the mechanisms at play.
Ans: We sincerely appreciate the reviewer’s thoughtful feedback on our experimental approach. We recognize the limitations of overexpression studies in fully capturing physiological relevance. However, due to constraints in available resources and feasibility (as it is challenging to obtain tissues that selectively express only D2R or β2AR), conducting additional experiments using endogenous protein levels or clinically relevant models was not feasible.
In future studies, we plan to conduct experiments using primary cells obtained from tissues using CRISPR Cas9 technology to control the expression of cellular components.
- This study does not mention the potential therapeutic implications or clinical relevance of the findings. The authors must, apart from that, speak to how their findings could be applied in models of disease or drug discovery, which will raise the general importance and relevance of the study.
Ans: We appreciate the reviewer’s valuable suggestion regarding the therapeutic implications and clinical relevance of our findings. In response, we have revised the discussion section to better highlight how our results could contribute to understanding disease mechanisms and potential therapeutic applications. Specifically, we have included a discussion on how our findings could be integrated into models of disease and drug discovery, providing a broader context for their significance.
GRK2 is implicated in the regulation of β-adrenergic receptors in the heart. Its overexpression has been associated with heart failure (Ferrero and Koch, 2022), suggesting that targeting GRK2 could be a therapeutic strategy. MDM2's Influence on GRK2 Activity: MDM2 has been shown to regulate cardiac contractility by inhibiting GRK2-mediated desensitization of β-adrenergic receptors (Jean-Charles et al., 2017), highlighting its potential as a therapeutic target in heart diseases.​
The endocytosis of b2AR serves as a potential pathway for receptor resensitization (Yu et al., 1993). Phosphorylation of D2Rs is critical for their recycling and resensitization, and disruption of this process leads to slower receptor recycling and enhanced desensitization (Cho et al., 2010). The critical importance of endocytosis in maintaining cellular homeostasis and its involvement in various diseases when dysregulated (Pathak et al., 2023).
- I did find some similarities between the manuscript and previously published work. I would recommend that the authors correct these concerns to avoid any ethical issues related to plagiarism. However, I do believe that the following remarks must be addressed before publication.
Ans: We sincerely appreciate the reviewer’s thorough evaluation and their diligence in identifying potential similarities between our manuscript and previously published work (PMID: 39273591).
While both articles delve into MDM2-mediated ubiquitination of GRK2 and arrestin3, they offer complementary insights: the older article focuses on the nuclear-cytosolic dynamics of these modifications, whereas the newer article emphasizes the impact of ubiquitination on endocytic pathway selection.
We take ethical concerns, including plagiarism, very seriously. To address this concern, we have thoroughly reviewed our manuscript and cross-referenced it with relevant literature to ensure that any overlapping content is properly cited and rephrased where needed, such as in the abstract.
In conclusion, although the study presents fascinating facts about regulating GRK2 and arrestin3 by Mdm2-mediated ubiquitination, the lack of robust experimental evidence and inadequate discussion of clinical relevance hinder its recommendation for publication in this highly regarded journal. The authors should address these concerns and provide additional experimental evidence before resubmission of the manuscript.
Ans: We sincerely appreciate the reviewer’s thoughtful feedback and recognition of the importance of our study in elucidating the regulation of GRK2 and arrestin3 through Mdm2-mediated ubiquitination.
The key finding of this paper is that non-ubiquitinated arrestin3 mediates a novel non-clathrin, non-caveolar endocytic pathway, which has not been previously characterized. To further validate this discovery, we tested our hypothesis by examining receptor endocytosis in the absence of GRK2 and the exclusive presence of non-ubiquitinated arrestin3. Additionally, we demonstrated that this third endocytic pathway functions in a dynamin-dependent manner (Fig. 5D).
Reviewer 2 Report
Comments and Suggestions for Authors
The manuscript by Wang et al. provides an in-depth analysis of the interplay between GRK2 and arrestin3 ubiquitination, highlighting their interdependence and their distinct roles in endocytic pathways. The study is comprehensive and provides valuable insights into the mechanisms of receptor endocytosis.
General comment:
The authors used a classical approach to assess receptor endocytosis by using hydrophilic radiolabelled ligands - [3H]-sulpiride D2R and [3H]-CGP12177 for β2-AR. While effective, this method does not provide real-time monitoring of endocytosis events. The inclusion of newer techniques (e.g. DERET-based assays) could improve the study by providing continuous real-time monitoring of receptor internalization.
Minor points:
- Although receptor constructs have previously been generated/characterized, it would be beneficial to specify that the short form of the D2R was likely used in this study.
- The authors propose an alternative clathrin- and caveolar-independent endocytic pathway mediated by non-ubiquitinated arrestin3. It needs to be clarified whether this pathway is dynamin-dependent or independent.
Author Response
Comments from Reviewer-2
The manuscript by Wang et al. provides an in-depth analysis of the interplay between GRK2 and arrestin3 ubiquitination, highlighting their interdependence and their distinct roles in endocytic pathways. The study is comprehensive and provides valuable insights into the mechanisms of receptor endocytosis.
General comment:
The authors used a classical approach to assess receptor endocytosis by using hydrophilic radiolabelled ligands - [3H]-sulpiride D2R and [3H]-CGP12177 for β2-AR. While effective, this method does not provide real-time monitoring of endocytosis events. The inclusion of newer techniques (e.g. DERET-based assays) could improve the study by providing continuous real-time monitoring of receptor internalization.
Ans: We appreciate the reviewer’s comments regarding the methodology used to assess receptor endocytosis. We acknowledge that real-time monitoring techniques, such as DERET-based assays, offer advantages in tracking dynamic internalization events. However, our choice of using hydrophilic radiolabeled ligands—[3H]-sulpiride for D2R and [3H]-CGP12177 for β2AR—was based on their well-established reliability in quantifying receptor internalization under the conditions of our study.
Currently, this experimental methodology has not been established in our laboratory, preventing its application in this submission. However, incorporating this method into our future studies could significantly enhance the depth and scope of our research.
Minor points:
Although receptor constructs have previously been generated/characterized, it would be beneficial to specify that the short form of the D2R was likely used in this study.
The authors propose an alternative clathrin- and caveolar-independent endocytic pathway mediated by non-ubiquitinated arrestin3. It needs to be clarified whether this pathway is dynamin-dependent or independent.
Ans: To address the reviewer’s comments, we performed a targeted experiment to determine whether receptor endocytosis occurs in a dynamin-dependent manner. Specifically, we tested this under experimental conditions where only non-ubiquitinated arrestin3, which mediates the third endocytic pathway, is functional. Our results demonstrate that this novel endocytic pathway relies on dynamin for efficient internalization (Fig. 5D).
Round 2
Reviewer 1 Report
Comments and Suggestions for Authors
Thank you so much for allowing me to review this revised article version. While I noticed that the suggested experiments were not included in this revision, I appreciate the author's thoughtful responses to the comments and detailed explanations addressing the concerns raised. The author's clarifications and the improvements made to the manuscript are commendable, and I believe the article is now in a strong position for publication. Based on authors' responses and the overall quality of the revised manuscript, I am satisfied with the revisions and am happy to accept the article for publication.